# Obinutuzumab-Induced B Cell Depletion Reduces Spinal Cord Pathology in a CD20 Double Transgenic Mouse Model of Multiple Sclerosis

**DOI:** 10.3390/ijms21186864

**Published:** 2020-09-18

**Authors:** Thomas Breakell, Sabine Tacke, Verena Schropp, Henrik Zetterberg, Kaj Blennow, Eduard Urich, Stefanie Kuerten

**Affiliations:** 1Institute of Anatomy and Cell Biology, Friedrich-Alexander-Universität Erlangen-Nürnberg (FAU), 91054 Erlangen, Germany; tom.breakell@fau.de (T.B.); sabine.tacke@fau.de (S.T.); verena.schropp@fau.de (V.S.); 2Department of Psychiatry and Neurochemistry, Institute of Neuroscience & Physiology, the Sahlgrenska Academy at the University of Gothenburg, 43141 Mölndal, Sweden; henrik.zetterberg@clinchem.gu.se (H.Z.); kaj.blennow@clinchem.gu.se (K.B.); 3Clinical Neurochemistry Laboratory, Sahlgrenska University Hospital, 43180 Mölndal, Sweden; 4Department of Neurodegenerative Disease, UCL Institute of Neurology, Queen Square, London WC1N 3BG, UK; 5UK Dementia Research Institute at UCL, London WC1E 6BT, UK; 6Roche Pharma Research and Early Development, Neuroscience, Roche Innovation Center, 4070 Basel, Switzerland; eduard.urich@roche.com

**Keywords:** anti-CD20, B cells, EAE, mAb, multiple sclerosis, neurodegeneration, obinutuzumab

## Abstract

B cell-depleting therapies have recently proven to be clinically highly successful in the treatment of multiple sclerosis (MS). This study aimed to determine the effects of the novel type II anti-human CD20 (huCD20) monoclonal antibody (mAb) obinutuzumab (OBZ) on spinal cord degeneration in a B cell-dependent mouse model of MS. Double transgenic huCD20xHIGR3 (CD20dbtg) mice, which express human CD20, were immunised with the myelin fusion protein MP4 to induce experimental autoimmune encephalomyelitis (EAE). Both light and electron microscopy were used to assess myelination and axonal pathology in mice treated with OBZ during chronic EAE. Furthermore, the effects of the already established murine anti-CD20 antibody 18B12 were assessed in C57BL/6 wild-type (wt) mice. In both models (18B12/wt and OBZ/CD20dbtg) anti-CD20 treatment significantly diminished the extent of spinal cord pathology. While 18B12 treatment mainly reduced the extent of axonal pathology, a significant decrease in demyelination and increase in remyelination were additionally observed in OBZ-treated mice. Hence, the data suggest that OBZ could have neuroprotective effects on the CNS, setting the drug apart from the currently available type I anti-CD20 antibodies.

## 1. Introduction

Multiple sclerosis (MS), a chronic inflammatory disease of the central nervous system (CNS) [1], manifests in dysfunction and apoptosis of oligodendrocytes and results in neurodegeneration [2]. It has a multifactorial etiology, with theories including genetics, viral causes, diet and vitamin D deficiency [3,4,5]. MS is the most common non-traumatic cause for neurological disability in young adults [6] with a worldwide prevalence of 2.3 million [7]. It has long been considered to be a predominantly T cell-mediated disease [8]. However, following the clinical success story of anti-CD20 antibody treatment in MS, B cells are attracting increasing attention regarding their role in the immunopathology of the illness [9]. Currently available and clinically developing antibodies, e.g., rituximab, ocrelizumab, ofatumumab and ublituximab are type I monoclonal antibodies (mAbs). While they are efficient at peripheral B cell depletion, primarily through induction of complement-dependent cytotoxicity (CDC) [10,11], they do not significantly deplete B cells within the CNS even when applied intrathecally. This is attributed to the rapid efflux of the drug from the CNS into peripheral blood [12].

The type II anti-CD20 mAb obinutuzumab (OBZ) is currently under investigation for the treatment of MS and is already approved for the treatment of follicular lymphoma and chronic lymphatic leukaemia [13,14].

Compared to type I, type II anti-CD20 mAbs are thought to deplete tissue-resident B cells more efficiently, using an Fc-effector independent direct B cell death inducing mechanism [10].

The aim of this study was to determine the effects of anti-CD20 antibody treatment on late-stage CNS histopathology in experimental autoimmune encephalomyelitis (EAE) [15], which is the most commonly used animal model of MS. In order to mirror the B cell component of MS pathology, we employed MP4-induced EAE, which relies on active disease induction with a fusion protein that consists of human myelin basic protein (MBP) and the three hydrophilic domains of proteolipid protein (PLP) [16,17]. We have previously demonstrated that MP4-induced EAE is both B cell- and antibody-dependent [18,19]. In this study two different models were used: Wild-type (wt) C57BL/6 (B6) mice were treated with the murine anti-mouse CD20 antibody 18B12 and huCD20xHIGR3 (CD20dbtg) mice received the humanised anti-CD20 antibody OBZ. CD20dbtg mice were created by cross-breeding the following two transgenic mouse strains: HuCD20tg mice that express both murine and human CD20 [20] and HIGR3 mice that express soluble human IgG since birth [21], resulting in their tolerance to it. Hence, the advantage of CD20dbtg mice is not only that they can be used to test therapeutic huCD20 mAbs such as the humanised OBZ, but also that they can be treated over a prolonged period of time since they will not generate a neutralising antibody response against the human drug.

Summarising our key findings, this study demonstrates a significant amelioration of spinal cord pathology using anti-CD20 treatment in both the wt and CD20dbtg model with slightly differential effects regarding axonal damage as well as de- and remyelination.

## 2. Results

### 2.1. Anti-CD20 Treatment Did Not Affect Clinical EAE Severity When Administred during the Chronic Stage of the Disease

In the wt cohort, *n* = 13 B6 mice were immunised with MP4, of which *n* = 7 were treated with 18B12 and *n* = 6 with muIgG2a starting on day 30 after the peak of EAE. In the CD20dbtg cohort 1, *n* = 12 CD20dbtg mice were immunised with MP4, of which *n* = 6 were treated with OBZ and *n* = 6 with huIgG1 starting on day 30 after the peak of EAE. In the CD20dbtg cohort 2, *n* = 12 CD20dbtg mice were immunised with MP4, of which *n* = 6 were treated with OBZ and *n* = 7 with huIgG1 starting on day 50 after the peak of EAE. Depletion efficacy of anti-CD20 treatment was verified in all cohorts using flow cytometry of blood samples on the day of sacrifice (Table 1).

As expected, MP4 induced chronic EAE in all mice. Clinical disease course and parameters for each cohort are shown in Figure 1 and Table 2. None of the cohorts showed a significant difference regarding the day of EAE onset, maximum EAE score, mean disease severity after onset and score at beginning of the treatment period when comparing the anti-CD20 and the isotype control antibody-treated groups. In addition, there was no significant difference in the EAE score before and after treatment in any of the three cohorts.

All cells were gated on single live lymphocytes. The percentage of the population of CD19^+^ B cells was related to the live population. The percentage of depleted cells was calculated by setting the isotype control at 100%. Values are displayed as mean values ± standard error of the mean (SEM). Statistical significance was determined using the Mann–Whitney *U* test.

### 2.2. 18B12 Treatment Significantly Reduced the Extent of Axonal Damage in Chronic MP4-Immunised Wt Mice

Wt mice were sacrificed seven days after final treatment with 18B12. In order to assess CNS pathology, the lesion area was measured in semi-thin sections of the spinal cord (Figure 2A) and ultrastructural analysis was performed by electron microscopy (Figure 2B).

As shown in Figure 2C, treatment with 18B12 significantly reduced the spinal cord lesion area in the ventrolateral tract (VLT) from 6.72% ± 1.84% to 0.49% ± 0.15%, which corresponds to a 92.8% decline compared to isotype control (*p* = 0.041). Similarly, ultrastructural analysis showed a reduction in the extent of axolysis from 25.43% ± 2.43% to 12.66% ± 2.75% in mice treated with 18B12, corresponding to a 50.2% decline compared to the isotype control group (*p* = 0.008). There was no significant effect on de- and remyelination.

### 2.3. OBZ Treatment Significantly Decreased Demyelination and Axonal Damage in Chronic MP4-Immunised CD20dbtg Mice

CD20dbtg mice were sacrificed seven days after final treatment with OBZ. In order to assess CNS pathology, the lesion area was defined in semi-thin sections of the spinal cord (Figure 3A) and ultrastructural analysis was performed by electron microscopy (Figure 3B).

Figure 3C demonstrates that treatment with OBZ starting on day 30 after the peak of EAE significantly reduced the spinal cord lesion area in the ventrolateral tract (VLT) from 9.14% ± 1.49% to 0.67% ± 0.22%, which corresponds to a 92.7% reduction compared to the isotype control (*p* = 0.004). Similarly, ultrastructural analysis showed a reduction of demyelination from 23.19% ± 4.52% to 11.82% ± 1.34%, corresponding to a 49.0% decrease (*p* = 0.022). There was also a significant reduction in the extent of axolysis (40.94% ± 6.33% in the isotype control group vs. 9.91% ± 2.24% in mice treated with OBZ, i.e., a reduction by 75.8% (*p* = 0.002)). No significant effect on remyelination was observed.

### 2.4. OBZ Treatment Significantly Increased Remyelination in Chronic MP4-Immunised CD20dbtg Mice

CD20dbtg mice were sacrificed seven days after final treatment with OBZ and spinal cord pathology was assessed as described in Section 2.3.

Ultrastructurally, treatment with OBZ starting on day 50 after the peak of EAE had no significant effect on lesion area or axolysis. However, there was a significant reduction in the extent of demyelination from 12.11% ± 1.99% to 0.17% ± 0.17%, which corresponds to a 98.6% decrease compared to isotype control (*p* = 0.002). Remyelination was increased from 0.57% ± 0.57% to 5.00% ± 1.46%—an increase of 779.8% (*p* = 0.009; Figure 4).

### 2.5. Serum NfL Levels Were Significantly Increased in EAE But Not Affected by Anti-CD20 Treatment in the Chronic Stage of the Disease

In order to broaden our point of view on neurodegeneration in the context of this study, we extended our analysis beyond electron microscopy, measuring serum levels of the neurofilament light (NfL) protein using ultrasensitive single molecule array (Simoa™) technology.

Neurofilaments (NFs) are intermediate filaments that are abundantly present in axons and responsible for their structural integrity. The NfL protein is currently being evaluated as a biomarker for neurodegeneration and disease progression in MS [22,23]. In Figure 5, the dashed line indicates the mean serum NfL level in non-immunised wt (*n* = 4) or CD20dbtg (*n* = 3) mice, respectively. Immunisation with MP4 triggered a significant increase in serum NfL levels (*p* = 0.01, Mann–Whitney *U* test, pooled values from all three cohorts). However, there was no significant difference between mice that received anti-CD20 antibody treatment and those treated with the corresponding isotype control antibody.

We correlated the degree of axolysis measured by electron microscopy, with serum NfL levels in individual mice. Figure 6 shows a fair correlation between these two parameters in all cohorts [24].

## 3. Discussion

In the past decades, research has mainly focused on the role of T cells in the pathogenesis of MS [8]. In search of new therapeutic options, this viewpoint has broadened and progressively shifted towards the contribution of B cells. While the role of pathogenic antibodies and their specificity in MS is subject to ongoing debate [25], there is accumulating evidence that B cells are primary antigen-presenting cells (APCs) that drive autoreactive T cells and enhance compartmentalised CNS responses [26]. Additionally, it has been shown that B cells are able to secrete both pro- and anti-inflammatory cytokines [27] and are involved in the development of ectopic follicles in the CNS. The latter were mainly found in patients with secondary progressive MS (SPMS), resulting in more severe disease progression including an earlier onset, more pronounced demyelination, microglial activation, axonal loss and increase in irreversible disability [28]. Further interest in the role of B cells has been fuelled by successful therapeutic targeting of the CD20 molecule [29].

Anti-CD20 antibodies are grouped into type I and type II mAbs, the only antibody approved for the treatment of MS being the type I mAb ocrelizumab [30]. Another type I anti-CD20 mAb, rituximab, failed to significantly deplete CNS-compartmentalised B cells regardless of whether it was applied systemically or intrathecally [12], subsequently leaving potential for improved therapeutic approaches.

OBZ is a novel humanised type II mAb that was developed for blood B cell malignancies and autoimmune diseases [31]. It is already approved for the treatment of follicular lymphoma [14] and chronic lymphatic leukaemia [13].

While OBZ neither relocalises CD20 into lipid rafts nor induces CDC as strongly as type I mAbs, it does exhibit increased induction of direct and immune effector cell-mediated cytotoxicity [32]. This results in OBZ exhibiting superior and more potent B cell depleting activity compared to rituximab in both mice and cynomolgus monkeys [10,33]. Another difference between type I and II anti-CD20 mAbs resulting in differing biological characteristics pertains to their epitope binding, that while being adjacent and partially overlapping, causes the antibodies to orientate differently upon binding [34]. Strikingly, OBZ reaches these superior effects despite displaying only half of the maximal binding to CD20 [35].

18B12 is a widely used agent for anti-CD20 targeted B cell depletion in mice: Barr et al. used it to ablate IL-6-producing B cells, which improved disease progression in EAE [36]. Yu at al. depleted B cells using 18B12 in a murine model of spontaneous autoimmune thyroiditis in young mice to show that the following activation of regulatory T cells inhibited development of the disease in adult mice [37]. Casili et al. showed that 18B12 treatment lessened inflammation and tissue damage in a model of spinal cord injury [38].

Here we used the B cell-dependent MP4-induced EAE model to study the effects of anti-CD20 treatment on chronic spinal cord pathology. We employed both 18B12 in wt B6 mice as well as OBZ in CD20dbtg mice. Our data demonstrate that both antibodies (18B12 and OBZ) were highly efficient at reducing axonal damage, while OBZ additionally reduced demyelination and induced late-stage remyelination. Apart from electron microscopic assessment, we also measured serum NfL levels in both EAE models using the highly sensitive Quanterix Simoa™ array [39]. NfL analysis is currently being evaluated as a biomarker in MS patients [22]. Traces of NfL become detectable in the peripheral blood after being shed into the cerebrospinal fluid (CSF) upon CNS tissue damage. Increased levels are suggested to indicate disease progression because they correlate with expanded disability status scale (EDSS) worsening and clinical relapses in some studies [40]. They were also reported to predict 10-year MRI brain lesion load (brain T2 hyperintense lesion volume) and atrophy (brain parenchymal fraction) in MS patients and were lowered accordingly under disease modifying treatment [23]. While NfL levels were elevated in MP4- versus non-immunised mice, we found only a fair correlation with axolysis in a pooled analysis of all three cohorts, and there was no significant impact of an anti-CD20 antibody treatment on NfL levels. It is conceivable that this lack of effect can be attributed to the late time point of measurement in our study. In EAE, axolysis occurs early on [41], is irreversible and reaches its maximum during acute disease before declining in chronic EAE, which is then characterised by axonal loss. Hence, to determine the optimal effects of anti-CD20 antibody application on NfL levels, treatment and NfL readout earlier in the disease course, during acute EAE, would be advisable. Along these lines, we also did not observe a beneficial effect of diminished spinal cord pathology on clinical disease outcome following anti-CD20 antibody treatment. Unless a treatment is applied that repairs damaged axons, no improvement in EAE score can be expected during the chronic stage of the disease.

The results of our study seem promising regarding a potentially beneficial impact of OBZ on CNS pathology. When interpreting these results, it is important to bear in mind that EAE mirrors only some aspects of MS and cannot equate to the disease in its entirety. EAE requires the application of pertussis toxin to facilitate blood–brain barrier permeabilisation and of myelin antigens to induce autoimmunity, whereas MS in humans develops through a multifactorial etiology [4]. While drugs often show promising results in animal models, they are not always effective in the treatment of MS, even worsening the disease or causing severe side effects [42]. Now, further work is needed to directly compare the therapeutic effects of OBZ with other anti-CD20 antibodies in the CD20dbtg mouse model and to explore the possible placement of OBZ within the MS treatment landscape.

## 4. Materials and Methods

### 4.1. Mice

Male B6 mice were purchased from Charles River (Sulzfeld, Germany) and male CD20dbtg mice were obtained from Taconic Biosciences (Silkeborg, Denmark). This transgenic mouse strain had been engineered to express both murine and human CD20, and to tolerate human IgG1 molecules (see above for details). All mice were eight to nine weeks old at the time of immunisation and kept under specific pathogen-free conditions at the animal facility (“Präklinisches Experimentelles Tierzentrum” (PETZ)) of the University Erlangen-Nürnberg. Humidity and temperature were maintained at 45–65% and 20–24 °C and the facility kept under a twelve-hour light–dark cycle. The mice had free access to a standard autoclaved rodent diet (Ssniff Spezialdiäten, Soest, Germany) and autoclaved tap water. Sufficient feeding and drinking for paralysed mice were ensured by supplying food and water in a gel form at ground level. Stocking density of the cages was determined according to EU guideline 2010-63.

All animal experiments were performed according to protocols that were approved by the “Regierung von Unterfranken” (ethical approval code: RUF-55.5.5-2532-2-577) and complied with the German Law on the Protection of Animals, the “Principles of Laboratory Animal Care” (NIH publication no. 86-23, revised 1985) and the “ARRIVE guidelines for reporting animal research” [43].

### 4.2. EAE Induction and Clinical Scoring

In preparation for EAE induction, complete Freund’s adjuvant (CFA) was obtained by combining nine parts of paraffin oil (Sigma-Aldrich, St. Louis, MO, USA) and one part of mannide monooleate (Sigma-Aldrich) with 5 mg/mL *Mycobacterium* tuberculosis H37 Ra (BD Difco Laboratories, Franklin Lakes, NJ, USA). All mice were subcutaneously immunised in both sides of the flank with a total of 200 µg MP4 (Alexion Pharmaceuticals, New Haven, CT, USA) emulsified in 200 µL CFA. On the day of immunisation and 24 h later, the mice received 100 ng pertussis toxin (Hooke Laboratories, Lawrence, MA, USA) in 100 µL sterile phosphate-buffered saline (PBS) intraperitoneally. Mice were scored daily to assess the progress of clinical symptoms according to the standard EAE scale: (0) no symptoms, (1) floppy tail, (2) hind limb weakness, (3) full hind limb paralysis, (4) quadriplegia and (5) death. Increments of 0.5 were used to account for intermediate scores not sufficiently determined by the five hallmarks.

### 4.3. Groups and Treatment

Mice were divided into three cohorts:“wt cohort” (wt);“CD20dbtg cohort 1” (CD20dbtg1);“CD20dbtg cohort 2” (CD20dbtg2).

The “wt cohort” consisted of *n* = 13 wildtype C57BL/6 mice, of which *n* = 6 were treated with a murine IgG-control antibody (muIgG2a; Bio X Cell, Lebanon, NH, USA) and *n* = 7 were treated with the murine anti-CD20 antibody 18B12 (Roche, Basel, Switzerland), both at a concentration of 5 mg/kg body weight in 50 µL PBS, into the lateral tail vein. Treatment was initiated at 30 days after the peak of EAE and repeated on day 33 and 36.

“CD20dbtg cohort 1” consisted of *n* = 12 transgene huCD20xHIGR3 mice, of which *n* = 6 were treated with a human IgG-control antibody (huIgG1; Roche) and *n* = 6 were treated with OBZ (Roche), both at a concentration of 5 mg/kg body weight in 50 µL PBS, into the lateral tail vein. Treatment time and frequency were equivalent to the wild-type cohort, initiated at 30 days after the peak of EAE and repeated on day 33 and 36.

“CD20dbtg cohort 2” consisted of the same strain and number of mice and received an equivalent treatment as “CD20dbtg cohort 1” but starting on day 50 after the peak of EAE and repeated on day 53, 56 and 59.

### 4.4. Tissue Sampling and Preparation

All mice were sacrificed with CO_2_ seven days after the final treatment. For Simoa™ NfL analysis, blood was collected from the inferior vena cava, centrifuged at 1000× *g* for 10 min and the serum stored at −80 °C until analysis. Mice were perfusion-fixed with 4% paraformaldehyde (PFA; Roth, Karlsruhe, Germany) in PBS. The lumbar spinal cord was dissected and post-fixed at 4 °C overnight using transmission electron microscopy (TEM) fixing solution under slight agitation. The TEM fixing solution was prepared by dissolving 4 g 4% PFA and 2.14 g 0.1 M sodium cacodylate (Serva, Heidelberg, Germany) in 50 mL double-distilled water (ddH_2_O). After cooling and filtration, 16 mL 25% glutaraldehyde (Roth) and 17 mL 1.2% picric acid (AppliChem, Darmstadt, Germany) were added and the solution was filled with ddH_2_O to reach 100 mL. The pH was titrated with either hydrochloric acid (Roth) or sodium hydroxide (Roth) to reach a pH of 7.2—7.4. The samples were then washed with PBS for 24 h, changing the buffer three times.

### 4.5. Light and Electron Microscopy

In preparation for light and electron microscopic analysis, spinal cord samples were treated with 1.5% potassium ferricyanide (Merck, Darmstadt, Germany) and 1% osmium tetroxide (Emsdiasum, Hatfield, PA, USA) in PBS for two hours at room temperature. In order to dehydrate and embed the samples, they were treated with 70% ethanol (Roth) for 60 min, then 80% ethanol, 90% ethanol, 100% ethanol, 100% ethanol (undenatured), 100% ethanol/acetone (Roth) 1/1, 2/3 acetone and 1/3 Epon, 1/3 acetone and 2/3 Epon for 45 min each and finally Epon and 2% glycidether accelerator DMP-30 (Roth) for 180 min. Epon was prepared by mixing two solutions (A and B) prior to use: Solution A consisted of 75 mL of glycidether 100 (Roth) and 120 mL of glycidether hardener DBA (Roth) and solution B consisted of 120 mL of glycidether 100 (Roth) and 105 mL of glycidether hardener MNA (Roth).

After dehydration, the samples were aligned in molds and coated with Epon and 2% glycidether accelerator DMP-30 and polymerised at 60 °C overnight.

For light microscopic analysis, the embedded tissue was transversely sliced at a thickness of 500 nm with an ultramicrotome (Leica Ultracut UCT, Wetzlar, Germany). Sections were covered in poly-l-lysine (Sigma, St. Louis, MO, USA) and transferred to a microscope slide to be dried at 80 °C on a heating plate before being colored at 70 °C with a Richardson’s methylene blue solution for 2–3 min. Richardson’s methylene blue was prepared by mixing two solutions (C and D) prior to use: Solution C consisted of 1% azur II (Merck) dissolved in 1% borax water (sodium tetraborate × 10H_2_O (Roth) in ddH_2_O) and solution D consisted of 1% methylene blue (Merck) in 1% borax water. After being rinsed with ddH_2_O and being left on the heating plate to dry, the sections were submerged in xylol (Roth) and then covered in DePex (Serva). Light microscopic images were taken on a Leica DM2000 microscope (Leica Microsystems, Heerbrugg, Switzerland) with a Leica MC190 HD camera (Leica Microsystems) and acquired using the Leica Application Suite X (LAS X) software (Leica Microsystems).

For electron microscopy, the Epon blocks were subsequently cut at a thickness of 80 nm, stretched with chloroform (Roth) and transferred to copper grids (Emsdiasum) before being contrasted with 10% uranyl acetate (Emsdiasum) for ten minutes, rinsed with ddH_2_O and then treated for another ten minutes with 2.8% lead(II)citrate × 3H_2_O (Polysciences Inc., Warrington, PA, USA). Electron micrographs were taken on a ZEISS EM 906 (Zeiss, Oberkochen, Germany) using a cathode voltage of 60 kV, a magnification of ×6000 and acquired using the iTEM Software (Olympus Soft Imaging Solutions, Münster, Germany).

Images were analysed using the Image J software (National Institutes of Health, Bethesda, MD, USA). Ten images per mouse were analysed, covering the same areas of the spinal cord in the respective treatment groups being compared, namely the area adjacent to the anterior fissure of the ventrolateral tract (VLT). All axons contained within these ten images were counted in order to detect the percentage of axolytic axons. This resulted in an average number of *n* = 218.86 ± 9.79 (mean value ± SEM) axons analysed per mouse. In order to calculate the g-ratio, we then randomly selected five of the ten images and measured 20 axons per image, thus a total of 100 axons per mouse were included into the g-ratio analysis. This is according to the standard procedure as also described in [44].

The experimenter was blinded to the identity of the treatments during analysis of the samples. Lesion area was defined by analysing a methylene blue stained cross section of the lumbar spinal cord, measuring the lesion area and dividing it by the area of the entire VLT.

In order to describe myelin pathology, axons were grouped into one of three categories: normal, demyelinating or remyelinating. This classification is based on the g-ratio, a concept that relates the diameter of an axon to the diameter of the axon including its myelin sheath by dividing the former with the latter [45]. Our analysis was focused on axons in the ongoing process of demyelination, which are initially characterised by a swollen disintegrating myelin sheath and hence a decreased g-ratio. When an axon remyelinates, the myelin sheath wraps around the axon slowly and successively, hence the g-ratio increases compared to a normally myelinated axon, as defined in [46]. In order to generate a quantitative statement, it is necessary to relate the single measured g-ratio values to a reference group of healthy animals. We used naïve/non-immunised mice of both the CD20dbtg (*n* = 3) and wt (*n* = 4) strains described above, which resulted in the following ranges for normal myelination: 0.65–0.78 for the CD20dbtg and 0.58–0.85 for the wt cohort. These values were obtained by the addition and subtraction of three standard deviations to the mean obtained from the reference mice. Additionally, axonal pathology was assessed by determining the percentage of axolytic axons. Exemplary demyelinating, remyelinating and axolytic axons are shown in Figure 7.

### 4.6. Flow Cytometry

One day before perfusion, blood was drawn from the tail vein and collected in Eppendorf tubes that contained 80 μL of heparin. Cells were stained with 1 µL of BD Horizon™ Fixable Viability Stain 450 (FVS450; BD Biosciences, San Jose, CA, USA) and 1 µL of APC anti-CD19 antibody (BioLegend, London, UK) at room temperature for 30 min. Erythrocytes were lysed with 3 mL 1× RBC lysis buffer (BioLegend) for 10 min followed by centrifugation at 500× *g* for 5 min. The supernatant was discarded and cells resuspended in 3 mL of FacsFlow (BD Biosciences). Cells were once again centrifuged and resuspended in 150 µL of FacsFlow. Flow cytometric acquisition was performed on a CytoFLEX S equipped with CytExpert 2.2. software (Beckman Coulter, Brea, CA, USA). Data analysis was performed using FlowJo version 10.0.6 (Tree Star Inc., Ashland, OR, USA). Dead cells were excluded from the analysis. Afterwards, doublets were excluded using the combined width parameter of the forward and side scatter.

### 4.7. Simoa™

Serum NfL levels were measured using ultrasensitive Single molecule array (Simoa™) technology, using the NF-light Advantage kit on an HD-X Analyzer according to instructions from the kit manufacturer (Quanterix, Billerica, MA, USA). The measurements were performed by board-certified laboratory technicians in one round of experiments using one batch of reagents. For a QC sample with a concentration of 7.4 pg/mL, the intra-assay coefficient of variation was 5.0%. For a QC sample with a concentration of 58.1 pg/mL, the intra-assay coefficient of variation was 4.0%.

### 4.8. Statistical Analysis

The Mann–Whitney *U* test was used to determine statistical significance and computed using GraphPad Prism 8 (San Diego, CA, USA). Statistical significance was set at *p* ≤ 0.05 and is visualised by the following symbols: *: *p* < 0.05; **: *p* < 0.01 throughout the manuscript. All graphs are displayed with mean values and standard errors of the mean (SEM).

## Figures and Tables

**Figure 1 ijms-21-06864-f001:**
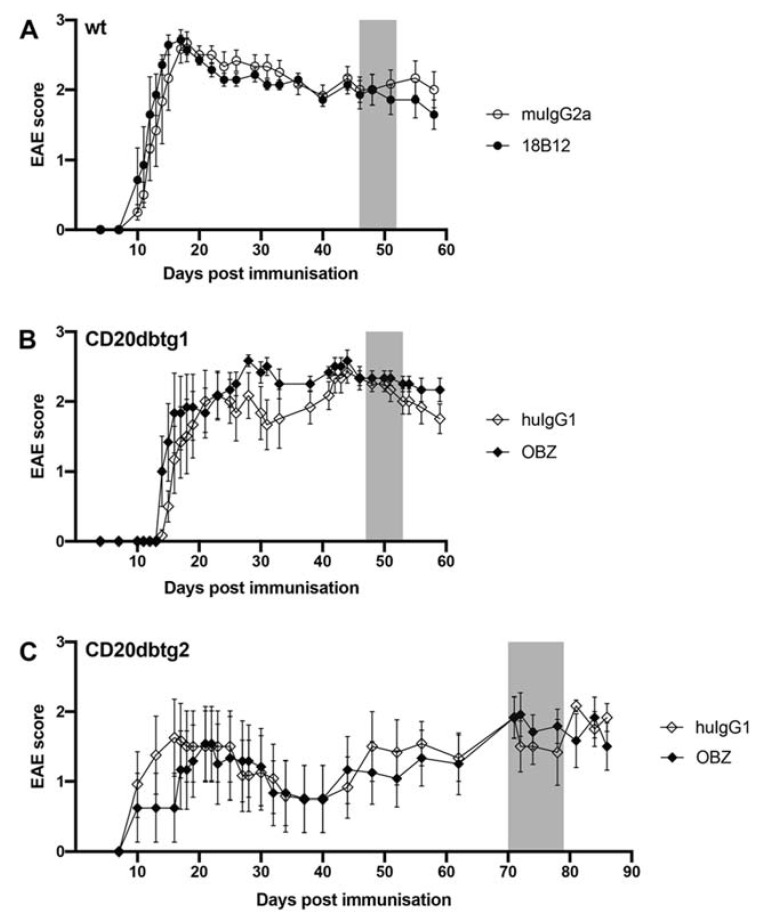
Disease course of mice treated with anti-CD20 or respective isotype control antibodies. (**A**) Wild-type (wt) cohort: Treated on day 30 after the peak of experimental autoimmune encephalomyelitis (EAE) with 18B12 or muIgG2a, respectively (*n* = 13). (**B**) CD20dbtg cohort 1: Treated on day 30 after the peak of EAE with OBZ or huIgG1, respectively (*n* = 12). (**C**) CD20dbtg cohort 2: Treated on day 50 after the peak of EAE with OBZ or huIgG1, respectively (*n* = 12). Treatment periods are marked in grey. Values are displayed as mean values ± standard error of the mean (SEM).

**Figure 2 ijms-21-06864-f002:**
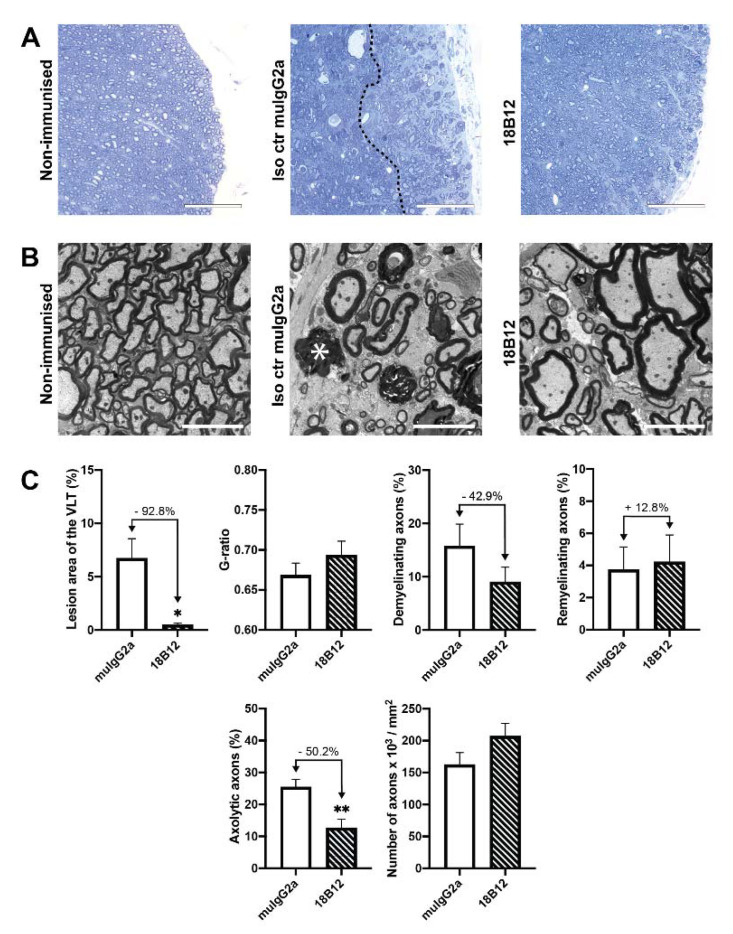
Spinal cord histopathology in wt mice treated from day 30 after the EAE peak with 18B12 or muIgG2a, respectively. (**A**) Representative light microscopic images of the spinal cord. The lesion area can be seen to the right of the dashed line. The scale bar represents 75 µm. (**B**) Representative electron micrographs of the spinal cord. The scale bar represents 5 µm. An exemplary axolytic axon is marked with a star. Images of untreated non-immunised mice have been added in both A and B for reference. (**C**) Quantification of the treatment effect on lesion area, g-ratio, demyelination, remyelination, axolysis and number of axons. Values are displayed as mean values ± standard error of the mean (SEM). *: *p* < 0.05; **: *p* < 0.01. Statistical significance was determined using the Mann–Whitney *U* test.

**Figure 3 ijms-21-06864-f003:**
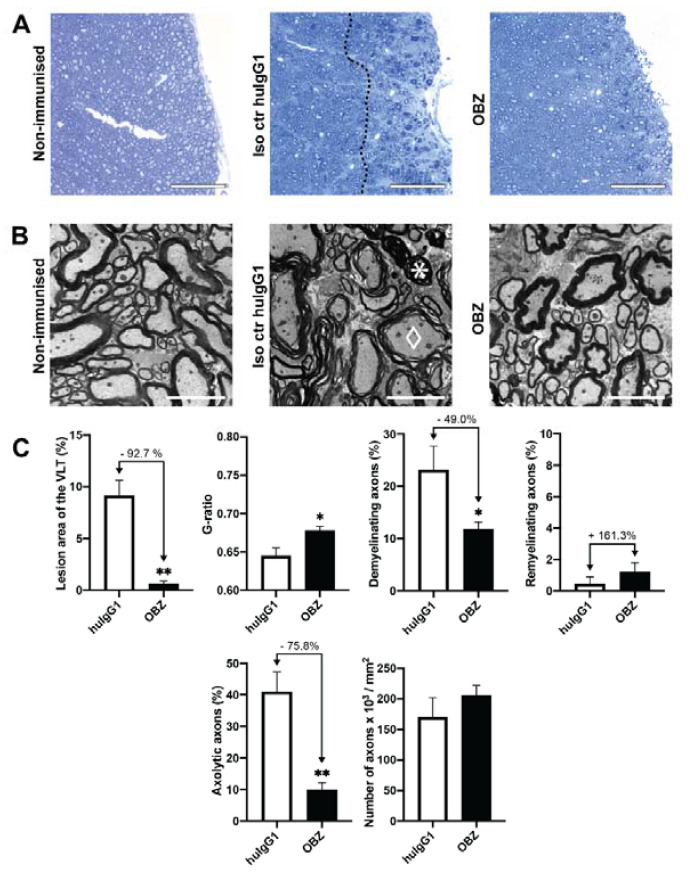
Spinal cord histopathology in CD20dbtg mice treated from day 30 after the EAE peak with OBZ or huIgG1, respectively. (**A**) Representative light microscopic images of the spinal cord. The lesion area can be seen to the right of the dashed line. The scale bar represents 75 µm. (**B**) Representative electron micrographs of the spinal cord. The scale bar represents 5 µm. A diamond marks an exemplary demyelinating axon and a star an axolytic axon, respectively. Images of untreated non-immunised mice have been added in both (**A**) and (**B**) for reference. (**C**) Quantification of the treatment effect on lesion area, g-ratio, demyelination, remyelination, axolysis and number of axons. Values are displayed as mean values ± standard error of the mean (SEM). *: *p* < 0.05; **: *p* < 0.01. Statistical significance was determined using the Mann–Whitney *U* test.

**Figure 4 ijms-21-06864-f004:**
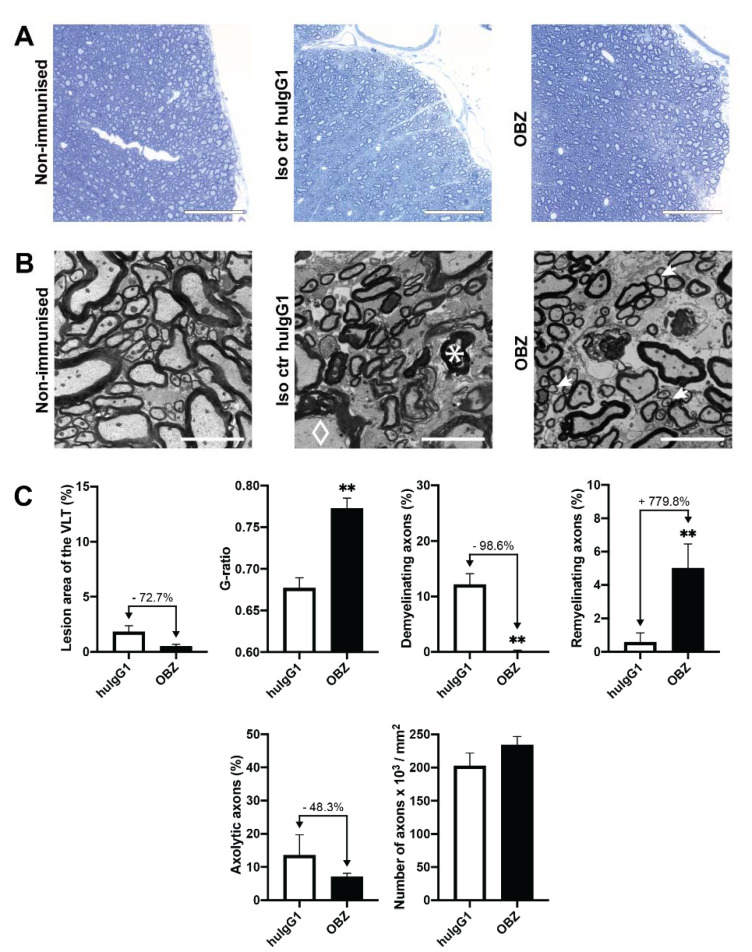
Spinal cord histopathology in CD20dbtg mice treated from day 50 after the EAE peak with OBZ or huIgG1, respectively. (**A**) Representative light microscopic images of the spinal cord. The scale bar represents 75 µm. (**B**) Representative electron micrographs of the spinal cord. The scale bar represents 5 µm. An exemplary demyelinating axon is marked with a diamond, three remyelinating axons with arrows and an axolytic axon with a star, respectively. Images of untreated non-immunised mice have been added in both (**A)** and (**B**) for reference. (**C**) Quantification of the treatment effect on the lesion area, g-ratio, demyelination, remyelination, axolysis and number of axons. Values are displayed as mean values ± standard error of the mean (SEM). *: *p* < 0.05; **: *p* < 0.01. Statistical significance was determined using the Mann–Whitney *U* test.

**Figure 5 ijms-21-06864-f005:**
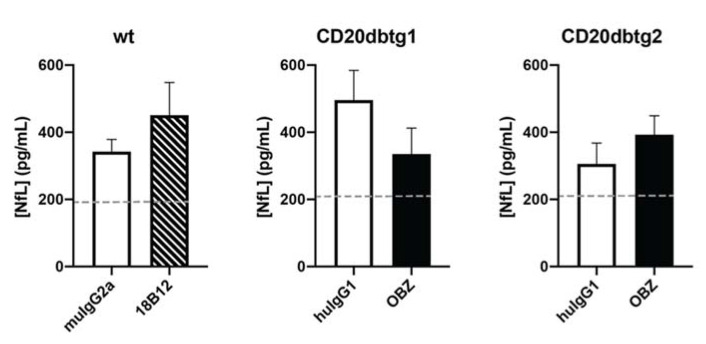
Serum NfL levels measured in each group (*n* = 6–7 mice per group). The dashed line indicates average values in untreated non-immunised mice (*n* = 3–4). Values are displayed as mean values ± standard error of the mean (SEM).

**Figure 6 ijms-21-06864-f006:**
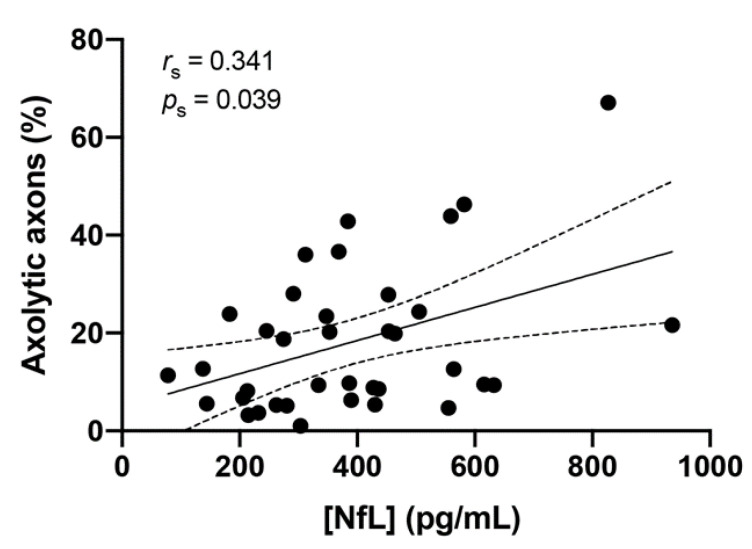
Correlation between the percentage of axolysis and serum neurofilament light levels in all mice from cohorts wt, CD20dbtg1 and CD20dbtg2 (*n* = 37). The regression line and 95% confidence intervals are shown.

**Figure 7 ijms-21-06864-f007:**
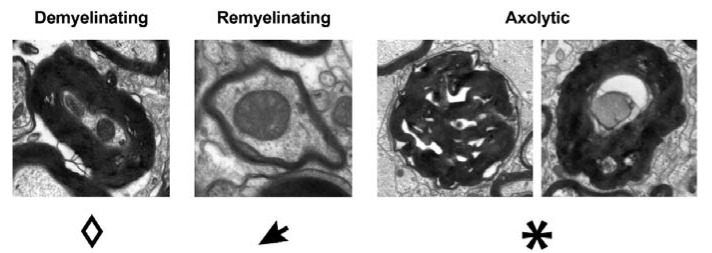
Exemplary images of axons in electron micrographs pertaining to the respective categories and the symbols used to mark them in the images in the results section.

**Table 1 ijms-21-06864-t001:** Depletion rates of CD19^+^ B cells in the peripheral blood of mice treated with anti-CD20 antibodies as determined by flow cytometry.

Cohort	Treatment	% of Depleted B Cells Compared to Isotype Control	*p* Value
wt	18B12	99.81% ± 0.08%	*p* < 0.001
CD20dbtg1	OBZ	97.13% ± 0.68%	*p* = 0.002
CD20dbtg2	OBZ	98.16% ± 0.33%	*p* < 0.001

**Table 2 ijms-21-06864-t002:** Clinical parameters of EAE in mice treated with anti-CD20 or respective isotype control antibodies.

Cohort	Treatment	Number of Mice	EAE Onset (Days after Immunisation)	Maximum Score	Mean Severity after Onset	Score at Beginning of Treatment	Score at End of Treatment	Score Difference before and after Treatment
**wt**	huIgG2a	6	12.17 ± 1.25	2.92 ± 0.08	2.20 ± 0.15	2.00 ± 0.18	2.00 ± 0.26	0.00 ± 0.13
	18B12	7	11.57 ± 0.48	2.93 ± 0.07	2.16 ± 0.09	1.93 ± 0.20	1.64 ± 0.21	−0.29 ± 0.10
*p* value			0.88	>0.99	>0.99	0.73	0.39	0.18
**CD20dbtg1**	huIgG1	6	16.67 ± 1.33	2.67 ± 0.17	1.97 ± 0.19	2.33 ± 0.17	1.75 ± 0.21	−0.58 ± 0.24
	OBZ	6	15.17 ± 0.65	2.75 ± 0.11	2.25 ± 0.13	2.33 ± 0.11	2.17 ± 0.17	−0.17 ± 0.11
*p* value			0.37	>0.99	0.59	>0.99	0.25	0.28
**CD20dbtg2**	huIgG1	6	22.67 ± 8.08	2.50 ± 0.18	1.65 ± 0.29	1.33 ± 0.33	1.92 ± 0.20	0.58 ± 0.40
	OBZ	6	23.33 ± 7.08	2.38 ± 0.18	1.46 ± 0.34	1.33 ± 0.40	1.50 ± 0.34	0.17 ± 0.38
*p* value			0.55	0.74	0.48	>0.99	0.40	0.47

Values are displayed as mean values ± standard error of the mean (SEM). Statistical significance was determined using the Mann–Whitney *U* test.

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
