# Peer review of "Obinutuzumab-Induced B Cell Depletion Reduces Spinal Cord Pathology in a CD20 Double Transgenic Mouse Model of Multiple Sclerosis"

_ijms, 2020, doi:10.3390/ijms21186864_

Round 1

Reviewer 1 Report

The paper by Breakell et al., examines the neuroprotective effects of OBZ in mouse MS models. Although overall the results are nicely presented, there are issues with some claims the authors are making that point to biased analysis. 1. G-ratios should be presented in all cases in the main figures. Furthermore, the authors are claiming that demyelinating axons are swollen and thus the the g-ratio is smaller than normal. The authors fail to acknowledge that in models of demyelination, myelin sheaths are thinner due to myelin loss. There is tons of literature that shows that the g-ratio is increased in demyelination, which the authors fail to show/discuss. Please present unbiased g-ratio data for all your experimental models. 2. There is no mention of the n number of axolytic, remyelinating and demyelinating axons the authors counted in the materials and methods. Usually, at least 100 (PMID: 24259565) to 300 axons (PMID: 29400711) per animal need to be analyzed and quantified. 3. The authors present data for axolytic axons (quantification and morphology). How about degenerated axons that are swollen and completely lack myelin as previously described (PMID: 29400711)? How about the total number of axons? 4. The authors should consider presenting data about the severity of white matter damage by using the Fazekas scale.

Author Response

G-ratios should be presented in all cases in the main figures.

Comprehensive graphs depicting g-ratios have been added to Figures 2-4 accordingly.

Furthermore, the authors are claiming that demyelinating axons are swollen and thus the the g-ratio is smaller than normal. The authors fail to acknowledge that in models of demyelination, myelin sheaths are thinner due to myelin loss. There is tons of literature that shows that the g-ratio is increased in demyelination, which the authors fail to show/discuss. Please present unbiased g-ratio data for all your experimental models.

We thank the reviewer for this comment. Our analysis was focused on axons in the ongoing process of demyelination, which are initially characterized by a swollen disintegrating myelin sheath and hence a decreased g-ratio. In our model, those axons clearly outnumbered axons that were almost completely demyelinated as characterized by an increased g-ratio. An increased g-ratio has been defined as a gold standard for unequivocally identifying remyelination (PMID: 25986556), and we applied this definition to our study. To accommodate the reviewer’s comment we have included this information on page 12 of the revised manuscript.

There is no mention of the n number of axolytic, remyelinating and demyelinating axons the authors counted in the materials and methods. Usually, at least 100 (PMID: 24259565) to 300 axons (PMID: 29400711) per animal need to be analyzed and quantified.

Following the reviewer’s comment, we have included this information on page 12 of the revised manuscript.

The authors present data for axolytic axons (quantification and morphology). How about degenerated axons that are swollen and completely lack myelin as previously described (PMID: 29400711)?

We thank the reviewer for this comment and have modified page 12 of the revised manuscript and Fig. 7 accordingly. We also checked PMID 29400711, specifically Figures 4 and S10 to better understand the reviewer’s definition of “degenerated axons that are swollen and completely lack myelin”. Indeed, this type of degeneration as referred to by the reviewer did not occur in our model to a degree that would permit reliable analysis which is why we omitted this category from our study.

How about the total number of axons?

We have now included graphs that show the total number of axons in Figures 2-4.

The authors should consider presenting data about the severity of white matter damage by using the Fazekas scale.

We would like to thank the reviewer for this suggestion. To the best of our knowledge the Fazekas scale is a radiological tool using MRI technology. Since our work was rather based on electron microscopy, we were not aware of the scale’s relevance to our study. We studied PMID: 29400711, in which the authors applied the Fazekas scale to LFB-stained sections of the corpus callosum. However, we believe it is not feasible to apply the modification of the original Fazekas scale as presented in this paper to our study. On the one hand, we used spinal cord sections in which axons were cut in transverse direction compared to the longitudinal direction of axons in the corpus callosum as in PMID: 29400711. On the other hand, we did not observe any vacuoles that are needed to define grade 2 of the scale.

Reviewer 2 Report

In this manuscript the authors compare the outcomes of different B cell depletion paradigms on the disease progression and spinal cord pathology in a model of EAE.  Transgenic animals expressing human CD20 develop a chronic inflammatory disease when immunized with MP4 myelin fusion proteins that results in chronic functional deficits. Comparison of the outcomes on disease progression and spinal cord pathology following treatment with anti CD20 antibodies were assessed in wild type mice treated with control and anti CD20 antibody 18B12 as well as transgenic animals treated with control and OBZ antibodies.  Animals were treated at 30- and 50-days post disease induction.  In no case were significant changes in disease progression detected through functional studies and all animals remained with a clinical score that was similar to pretreatment levels. Analysis of the pathology of the spinal cord by Toluidine blue and Em analyses suggested that treatment of wildtype animals with 18B12 reduced lesion load compared to controls while treatment with OBZ reduced spinal cord pathology in transgenic animals compared to controls. Based on EM studies the authors also suggest that treatment with OBZ increased remyelination in MP4 immunized transgenic animals

There are some interesting aspects to this manuscript, however there are some concerns with the data and its interpretation.

While the observation that treatment with antiCD20 antibodies at the time selected did not affect disease progress in any cohort, it is unclear why the level of disease is so different between the different cohorts of animals and suggest there is an inconsistency in the model which may influence data analysis.

If the claim is that OBZ is more effective than other CD20 antibodies it would be important to test that comparison directly and this appears to be missing.

The weakest part of the story is the suggestion that OBZ promotes remyelination. The differences in pathology between the models is relatively clear, but the EM data suggesting remyelination is very weak.  The region selected in Fig 4B does not appear to be in a lesion and the normal spinal cord contains axons that look similar to those indicated. Without much more convincing data (earlier sacrifice time, g ratios etc.) the authors should significantly deemphasize this aspect of the work.

The NFL data appears poorly connected to the rest of the paper in the current version.

One of the most interesting aspects of this paper is that although there are significant changes in the pathology of the spinal cord following anti CD20 treatment this is not reflected in the functional outcomes. A further discussion of this observation would make the current manuscript more timely.

Author Response

While the observation that treatment with antiCD20 antibodies at the time selected did not affect disease progress in any cohort, it is unclear why the level of disease is so different between the different cohorts of animals and suggest there is an inconsistency in the model which may influence data analysis.

It is true that the level of disease was slightly lower in the CD20dbtg cohort 2. However, statistical analysis resulted in no statistically significant difference for maximum score and mean severity after onset between any of the groups as determined by Kruskal-Wallis test and Dunn’s post-hoc analysis. It should also be noted that the lower score of the CD20dbtg cohort 2 at the beginning of treatment can be explained by the later time point of drug application (day 50 vs. day 30 in the other two cohorts).

If the claim is that OBZ is more effective than other CD20 antibodies it would be important to test that comparison directly and this appears to be missing.

We thank the reviewer for this highly valid comment and reasonable suggestion. Unfortunately, the number of CD20dbtg mice available for our study was very restricted and such mice are no longer available at this point in time. This is why we cannot perform any further experiments to test other anti-CD20 antibodies. However, to accommodate the reviewer’s comment we have extended the discussion on page 10 of the revised manuscript accordingly.

The weakest part of the story is the suggestion that OBZ promotes remyelination. The differences in pathology between the models is relatively clear, but the EM data suggesting remyelination is very weak.  The region selected in Fig 4B does not appear to be in a lesion and the normal spinal cord contains axons that look similar to those indicated. Without much more convincing data (earlier sacrifice time, g ratios etc.) the authors should significantly deemphasize this aspect of the work.

We thank the reviewer for this comment. We have checked Fig. 4B and exchanged the image for a more representative one. In addition, also following a comment made by reviewer #2 we are now including graphical representations of the g-ratio. We do not believe that an earlier sacrifice time point would have been feasible since remyelination mainly occurs late in the EAE model (PMID: 23899992).

The NFL data appears poorly connected to the rest of the paper in the current version.

We have added a bridging sentence on page 8 of the revised manuscript, to better connect the NfL data to the rest of the paper.

One of the most interesting aspects of this paper is that although there are significant changes in the pathology of the spinal cord following anti CD20 treatment this is not reflected in the functional outcomes. A further discussion of this observation would make the current manuscript more timely.

The reason for a missing beneficial effect of improved spinal cord pathology on functional outcomes might reside in the fact that treatment was initiated during chronic EAE, i.e. after the peak of disease. In EAE, axonal damage occurs early on in the disease (PMID: 23899992). Since it is irreversible in its nature, the functional deficits resulting from axonal damage are also irreversible. Hence, unless a treatment is applied that repairs damaged axons, no improvement in EAE score can be expected in the chronic stage of the disease. We have extended our discussion accordingly on page 10 of the revised manuscript.

Reviewer 3 Report

The manuscript is fine and very interesting in the topic of MS.

A few  amendments should be done:

  1. Table 2 subtitles are with a 't' letter away from each column tops
  2. Figure 2: English: change part of caption as ...' an example of axolysis is marked by a star...
  3. Figure 3: a diamond mark labeled a demyelinated axon ...

Author Response

The manuscript is fine and very interesting in the topic of MS. A few amendments should be done:

  1. Table 2 subtitles are with a 't' letter away from each column tops
  2. Figure 2: English: change part of caption as ...' an example of axolysis is marked by a star...
  3. Figure 3: a diamond mark labeled a demyelinated axon ...

We would like to thank the reviewer for the positive feedback and the amendment suggestions, which we have integrated accordingly.

Round 2

Reviewer 1 Report

No additional comments

Reviewer 2 Report

The authors have responded effectively to the previous round of reviews and the paper is improved. I have no further comments.